# Impact of High-Pressure Homogenization on Enhancing the Extractability of Phytochemicals from Agri-Food Residues

**DOI:** 10.3390/molecules28155657

**Published:** 2023-07-26

**Authors:** Annachiara Pirozzi, Francesco Donsì

**Affiliations:** 1Department of Industrial Engineering, University of Salerno, Via Giovanni Paolo II, 132, 84084 Fisciano, Italy; apirozzi@unisa.it; 2ProdAl Scarl, University of Salerno, Via Giovanni Paolo II, 132, 84084 Fisciano, Italy

**Keywords:** agri-food waste, bioactive compounds, extraction, high-pressure homogenization, circular economy

## Abstract

The primary objective of the Sustainable Development Goals is to reduce food waste by employing various strategies, including the reuse of agri-food residues that are abundantly available and the complete use of their valuable compounds. This study explores the application of high-pressure homogenization (HPH), an innovative nonthermal and green treatment, for the recovery of bioactive compounds from agri-food residues. The results demonstrate that the optimized HPH treatment offers advantages over conventional solid/liquid extraction (SLE), including shorter extraction time, solvent-free operation, low temperatures, and higher yields of phenol extraction (an approximately 20% improvement). Moreover, the micronization of agri-food residue-in-water suspensions results in a decrease in the size distribution to below the visual detection limit, achieved by disrupting the individual plant cells, thus enhancing suspension stability against sedimentation. These findings highlight the potential of HPH for environmentally friendly and efficient extraction processes.

## 1. Introduction

Agri-food industries create high volumes of organic waste (biomass), reaching up to 140 billion tons per year [1]. While a considerable portion of this waste is unrelated to food wastage issues [2], its disposal poses costs for food processors and has a negative impact on the environment. However, recovering value-added compounds, such as polyphenols, anthocyanins, tannins, and vitamins from agri-food residues represents a formidable challenge [3], as this can enhance the sustainability of food industry production, create economic and social benefits, and mitigate environmental burdens [4]. The concept of the zero-waste economy advocates for utilizing waste as a raw material for novel food products, ingredients, and applications [5]. Figure 1 schematically illustrates the recovery of bioactive fractions through conventional and novel technologies and their consequent valorization as natural additives or active ingredients in various sectors [6], including food, cosmetics, and pharmaceuticals [7].

Conventional solid/liquid extraction (SLE) techniques have been widely employed for extracting valuable compounds from food wastes/byproducts due to their ease of operability and scalability. However, SLE is hindered by the presence of the cell envelopes (membranes and walls) in plant tissues, which exert a significant resistance to the mass transfer of solvents and target intracellular compounds, thereby slowing down the solvent extraction process. The extraction yield can be increased by using specific solvents with very high affinities for the compounds to be extracted, which are generally toxic and harmful. Additionally, due to the incomplete removal of solvents in downstream processes, the extracted phytochemicals may have adverse effects on human health, potentially impacting the quality of the extract (for example, losing the functionality of the desired compounds) and, consequentially, their applications. The use of green and food-grade solvents, which have lower affinities for their target compounds, requires longer extraction times, high-purity solvents, and heat treatments, resulting in the degradation of thermolabile phytochemicals [5,6,7]. Therefore, efforts have been directed toward developing more efficient, sustainable, and ecologically friendly extraction technologies, such as enzyme-assisted extraction [8,9], supercritical fluid extraction, microwave-assisted extraction [10], and ultrasound-assisted extraction [11]. These extraction methods have thus far mainly been used to induce the permeabilization, lysis, or disintegration of the cell envelopes.

In recent years, several innovative and alternative approaches, such as pulsed electric fields (PEF) [12], high-pressure homogenization (HPH) [4], supercritical fluid extraction (SFE) [13], and ultrasound-assisted extraction (US) [14], have emerged as promising approaches for the extraction and recovery of bioactive compounds from plant materials, offering enhanced efficiency and improved product quality. This study explores high-pressure homogenization (HPH) technology, a nonthermal and easily scalable cell disruption technique for plant tissues involving solvent-free extraction methods. HPH induces the rupture of cell envelopes, enhancing (i) the extractability of target intracellular compounds from food byproduct matrices (for example, polysaccharides from large-leaf yellow tea [15]); (ii) the physicochemical and functional properties of dietary fiber [16]; and (iii) the functional properties of novel proteins [17]. Importantly, HPH achieves this without the need for organic solvents, ensuring the safety of the final products intended for use in food supplements and pharmaceutical and cosmeceutical applications. However, limited studies have demonstrated the feasibility of HPH in enhancing the extractability of phenolic compounds from agri-food residues.

To fully exploit the potential benefits of HPH-assisted extraction over conventional SLE process, optimization is required. Response surface methodology (RSM) has been successfully applied to optimize SLE conditions by evaluating the effects of various organic solvents and concentrations on total phenolic content. The primary goal is to determine the minimum treatment severity that maximizes the extraction yield. In this work, the selected solvents and concentration for the SLE process are used to determine the optimal extraction time through comprehensive kinetic studies. To gain further insight, the composition of polyphenols in the obtained extracts and their antioxidant properties are evaluated using HPLC-PDA analysis and FRAP analysis, respectively. Building upon this foundation, the study is extended to HPH-assisted extraction, employing water as the process medium on various agri-food residues. The specific focus lies in optimizing the treatment temperature for this novel extraction approach. The impact of HPH-assisted extraction is thoroughly evaluated by assessing the extraction yields of total phenols and antioxidant activity by Folin–Ciocalteau and FRAP assays, respectively. Additionally, the morphological and size properties of the micronized samples are analyzed to assess the physical effects of HPH-assisted extraction.

## 2. Results and Discussion

### 2.1. Characterization of Agri-Food Residues

The characteristics of the raw materials used in the present study are presented in Table 1.

The moisture, ash, lipid, protein, and carbohydrate contents, determined using official analytical methods, align with values reported in the literature [3,9,10,11,12,13,14]. Slight differences in the chemical composition between this study and the literature can be attributed to variations in climate and soil conditions, crop variety, agricultural practices, post-harvest management, and the genetic characteristics of the analyzed samples.

### 2.2. Conventional SLE Extraction Process

Hemp cake and sunflower cake, which are byproducts of oilseed cakes and sunflower oil refining, respectively, are known as rich sources of polyphenols, which are mainly located in the hull [18,19,20]. Other agri-food residues, such as spent coffee grounds from the industrial production of soluble coffee [21,22], lignocellulosic materials such as wheat bran, wheat middlings [23,24], and rice husk [25], as well as other wastes from the wood industry, have been reported to be good sources of phenolic compounds, which are obtained through the hydrothermal and/or autohydrolysis processing of lignin [1]. Additionally, the main byproducts of the tomato processing and wine industries, such as tomato and grape pomace, contain various phenolic compounds, such as flavanones and flavonols, quercetin, rutin, and kaempferol glycoside derivatives [26,27,28]. These agri-food residues still retain valuable polyphenols with health-beneficial properties, including cardioprotective, neuroprotective, anti-inflammatory, anticarcinogenic, antioxidant, and antimicrobial activities [28]. The SLE extraction process was carried out using different organic solvents at appropriate ratios of solvent–water mixtures, with the details given in the Appendix A. The optimal extraction conditions were determined through response surface methodology (RSM), taking into account (i) the experimental values of total phenols and the predicted values, (ii) the interactions of total phenols in the response variable for each solvent and different agri-food residues (Appendix A), and (iii) the evaluation of the fitting regression coefficients and analysis of variance (Appendix A, respectively) of the second-order polynomial model. Furthermore, based on the kinetic studies (Appendix A), the extraction time was set at 24 h under the optimized conditions (reported in Appendix A). Subsequently, the total phenol contents of the extracts of different agri-food residues were determined (Figure 2).

All agri-food residues contained substantial amounts of total phenolics, ranging from approximately 2 to 42 mg_GAE_/g_DM_. These values are consistent with those reported in the literature, with roasted coffee beans exhibiting higher total phenols than other agri-food residues [22], such as hemp cake [20], sunflower cake [19], grape pomace [29], tomato pomace [30], and lignocellulosic byproducts [31], which had values below 20 mg mg_GAE_/g_DM_. These results support the notion that agri-food residues can serve as valuable sources of phenolic compounds. Furthermore, a qualitative identification of phenolic compounds in extract form was conducted using HPLC analysis with a UV-Vis diode array detector (PDA) (Figure 3).

Chlorogenic acid and quercetin-3-glucoside were the predominant phenolic acids in all agri-food residues. The antioxidant capacity of the extracts was evaluated using ferric-reducing antioxidant power and inhibition percentage, which were assessed by FRAP (Figure 4a) and DPPH assays (Figure 4b), respectively.

The organic solvent concentration positively influenced the FRAP response (Figure 4a), indicating a higher concentration of hydrophobic phenols, which possess strong reducing properties and contribute to the antioxidant capacity of the extracts. In contrast, the DPPH radical response (Figure 4b) was negatively affected. The positive effect of the organic solvent concentration on the FRAP response and the low response to DPPH radicals could be due to the low ability to reduce Fe^3+^, suggesting a higher concentration of hydrophobic phenols [32]. The correlation between DPPH and the total phenolic content of the extracts was not significant (*p* > 0.05). However, the antioxidant activity measured by FRAP showed a strong positive correlation with the total phenol content, suggesting that phenolic compounds were primarily responsible for the extracts’ antioxidant capacity [33]. Notably, extracts derived from roasted coffee beans exhibited higher antioxidant activity compared to other extracts, which was likely due to the higher amount of recovered phenolic compounds. The antioxidant capacities of different agri-food residues, as reported by previous researchers, are in support of our results [34,35,36,37]. Nevertheless, since not all phenolic compounds show antiradical capabilities, the correlation between the polyphenols found in agri-food residues’ extracts and their reducing antioxidant capacity (FRAP) must be carefully analyzed and needs further studies. Thus, our analysis investigated the relationship between the total phenols and the FRAP-reducing activity of extract samples obtained under different extraction conditions. We thus aimed to elucidate the effect of the extracts’ total phenols on their in vitro antioxidant activity. The results revealed a strong positive correlation between the total phenol contents and their antioxidant capacity, which could be described by means of a linear function (y=0.852·x, R^2^ = 0.929) with a Pearson correlation coefficient of about 0.954. These findings indicate that the total phenol contents of agri-food residue extracts contribute significantly to the extracts’ ability to donate hydrogen electron [38]. In other words, the overall antioxidant activity of the extracts is mostly associated with the presence of phenolic compounds.

### 2.3. HPH-Assisted Extraction Process

#### 2.3.1. Effect on Bioactive Compound Extraction

Conventionally, phytochemicals are recovered from plant materials using SLE techniques. This study investigated an alternative approach based on the use of HPH-assisted extraction with water as the sole solvent. The aim was to investigate whether HPH, through its mechanical and nonthermal effects, could enhance the extraction of phytochemicals from various agri-food residues. The main working hypothesis was that HPH could improve the extractability of intracellular compounds, such as phytochemicals, by disrupting cell structures and facilitating their release. Different operating parameters were assessed to evaluate the effect of extraction temperature on total phenol extraction (Figure 5a) and the extracts’ antioxidant-reducing activity (Figure 5b). The HPH process involved passing the agri-food residue suspensions through a small orifice with a diameter of 200 µm under high pressure (80 MPa). However, certain agri-food residues with high lignocellulosic material content and rigid structures, such as hemp and sunflower cake, grape pomace, and rice husk, tended to agglomerate and form larger masses, leading to the clogging of the orifice and preventing the materials from going through the HPH treatment chamber. As a result, these particular raw materials could not be effectively treated using the HPH-assisted extraction method. Therefore, only aqueous suspensions of roasted coffee beans, wheat middlings, wheat bran, and tomato pomace were processed by HPH.

It was found that HPH-assisted extraction at 80 MPa and 5 °C for 20 equivalent passes, using water as the extraction medium, resulted in increased phenol extraction yields (Appendix A) compared to the conventional SLE process with the optimized organic solvent (Appendix A). Specifically, for roasted coffee beans, wheat middlings, wheat bran, and tomato pomace, the phenol extraction yield increased by 13%, 21%, 23%, and 19%, respectively. The temperature of the HPH-assisted extraction process was also observed to have a significant influence on the extraction of polyphenols. HPH-assisted extraction at room temperature (25 °C) further improved the extraction yield by an average of 21%. Generally, higher temperatures have a positive impact on the extraction of phenolic compounds from vegetal sources [39,40] by enhancing the solubility of polyphenols, increasing the diffusivities of the extracted molecules, and improving mass transfer [41]. However, it should be noted that excessively high temperatures (around 50 °C) may result in the release of certain phenolic compounds while potentially causing the thermal oxidation or degradation of others [40]. For example, when Rajha et al. (2013) compared the water extraction yield of phenolic compounds from grape pomace (Cabernet Sauvignon variety) in a range of temperatures between 33 and 50 °C, they found that diffusion time had a negative quadratic effect on phenols, showing a maximum total phenolic content at 30 h [42]. Longer extraction times had a negative impact on phenolic compounds, likely due to oxidation or degradation reactions triggered by the oxygen present in the headspace of the test flasks, which were accelerated at higher temperatures [42]. Moreover, it is worth noting that, regardless of the agri-food residue used, the positive correlation between the phenol concentration and reducing activity leads to a Pearson correlation coefficient higher than 0.999, indicating that the observed antioxidant activity is completely caused by the extracted phenolic compounds.

The extraction of phenolic compounds from HPH-treated agri-food residues and the exposure of antioxidant functional groups, which improve antioxidant capacity, are strongly related to the fluid-mechanical stresses in the homogenization valve [43,44]. They contribute to effectively increasing the specific surface area of vegetable materials through size reduction and, more importantly, loosening their microstructure, exposing more functional groups to the surrounding liquid phase and even creating pores or cavities inside them [45].

#### 2.3.2. Effect on Physical Characteristics of Agri-Food Residue Suspensions

The mechanical treatments of high-speed mixing (HSM) and HPH induced noticeable disruptions of the vegetable tissue, which was more evident for HPH. Visual observations showed that agri-food residue suspensions became more homogeneous in appearance after HPH treatment compared to samples treated with HSM, as depicted in Figure 6. Microscopic observations (Figure 7) supported these findings, indicating that HSM caused only fragmentation of the tissue into smaller cell aggregates, with minimal effects on cell wall integrity. On the other hand, HPH treatment resulted in more individual cell disruption, as evidenced by the presence of filamentous debris in the suspension, suggesting cell wall breakage. The variations in the sizes and shapes of the samples before and after HPH treatment were attributed to the fluid–mechanical stresses exerted during the treatment, including the shear stress, elongation, hydrodynamic cavitation, turbulence, and pressure gradient. These stresses improved the trimming of the particles along their length. Similar results were reported by Jurić et al. (2019) for tomato peels [45] and Wang et al. (2013) for wheat bran [46], supporting the findings of the present study.

The extent of cell disruption and size reduction was reflected in the particle size distribution curves reported in Figure 8 and the characteristic diameters (d(0.1), d(0.5), d(0.9), D[4,3], and D[3,2]) reported in Table 2.

Both mechanical methods, HSM and HPH, led to smaller particle sizes compared to non-micronized samples. Notably, the smallest particle sizes were observed when HPH was applied to the agri-food residue-in-water suspensions.

Specifically, HPH treatment significantly reduced the diameters corresponding to the 10th, 50th, and 90th percentile of the cumulative distribution (d(0.1), d(0.5), d(0.9). The distribution width, expressed as d(0.9)–d(0.1), ranged between 580 and 1020 μm for HSM and between 55 and 90 μm for HPH. Moreover, the volumetrically weighted mean diameters (D[4,3]) decreased by approximately 94%, 88%, 89%, and 92% for roasted coffee beans, wheat middlings, wheat bran, and tomato pomace, respectively. The surface-weighted mean diameters (D[3,2]) exhibited a change of around 78%, 52%, 75%, and 87% for roasted coffee beans, wheat middlings, wheat bran, and tomato pomace, respectively, which was slightly less than D[4,3]. Although micronizing is more challenging [47,48], HPH treatment enabled a decrease in both D[4,3] (influenced by larger particles) and D[3,2] (influenced by smaller particles). The variation in particle size also influenced the stability characteristics of the agri-food residue-in-water suspensions. Particles with larger dimensions tend to settle, while smaller particles exhibit increased interparticle interactions, resulting in the formation of a well-connected sample network that enhances suspension stability. This network restricts particle movement and reduces sedimentation, thereby maintaining the suspension’s stability. By altering particle size, it was thus possible to modify the stability of agri-food residues’ suspensions in water [49,50].

Remarkably, HSM treatment was not capable of destroying individual plant cells, whereas HPH treatment achieved complete disruption of the plant cells. This increased the suspension’s stability against sedimentation and facilitated the release of most intracellular contents into the suspension. These findings align with the results of the analysis of the bioactive compounds released in water upon micronization treatment, indicating the enhanced bioaccessibility of the antioxidant compounds typically present inside the cells.

## 3. Materials and Methods

The composition of raw materials and the bioactivity of high-value-added compounds extracted were determined by applying the official analytical methods described in the following sections.

### 3.1. Raw Materials

All agri-food residues, which were collected as raw material from a local milling industry (Salerno, Italy), were dry residues (with less than 10% moisture in the dry matter). They were stored as received, except for the tomato and grape pomaces, which were stored in aluminium trays and frozen at −20 °C until used.

### 3.2. Chemicals

The reagents used in this study were sulfuric acid (H_2_SO_4_, 98%, Sigma-Aldrich, St. Louis, MO, USA), copper(II) sulfate pentahydrate (CuSO_4_·5H_2_O, ACS Reagent, Fisher Scientific, Waltham, MA, USA), potassium phosphate monobasic (K2SO4, ≥99.5%, Honeywell Fluka Charlotte, NC, USA), 40 wt% sodium hydroxide solution (NaOH, PanReac, Barcelona, Spain), chloridric acid (HCl, 36.5–38.0%, ACS Reagent, PanReac), ethanol (C_2_H_5_OH, 99.9%, VWR Chemicals, Radnor, PA, USA), petroleum ether (C_6_H_14_, ACS Reagent, Sigma-Aldrich), diethyl ether ((C_2_H_5_)_2_O, ≥98%, VWR Chemicals), Folin–Ciocalteau’s reagent (VWR Chemicals), sodium carbonate anhydrous (Na_2_CO_3_, ≥99.5%, ACS Reagent, PanReac), phosphoric acid (H_3_PO_4_, 85–90%, Honeywell Fluka Charlotte), methanol (CH_3_OH, ≥99.8%, VWR Chemicals), 2,4,6-Tris (2-pyridyl)-s-triazine (C_18_H_12_N_6_, 99%, Fisher Scientific), iron (III) chloride hexahydrate (FeCl_3_·6H_2_O, ≥98%, Sigma-Aldrich), and 1,1-diphenyl-2-picryl hydrazyl (C_18_H_12_N_5_O_6_, Sigma-Aldrich). The water used was purified by a Milli-Q water purification system (Barnstead Pacific TII Water, Thermo Scientific, Waltham, MA, USA).

### 3.3. Proximate Composition Analysis of Agri-Food Residues

The moisture and ash contents of the samples were gravimetrically determined by drying at 105 °C in a forced-air oven (AOAC Method 950.46) [51] and at 525 °C in a muffle (AOAC Method 923.02) [51], respectively. The dry mass content was expressed as g of dry matter per g of sample (g_DM_/g), while the ash content was expressed as g of ashes per g of dry sample (g/g_DM_).

The total protein content was determined by the Kjeldahl method, following the procedures described by the Association of Official Analytical Chemists International, as indicated above (AOAC Method 954.01) [51], using a traditional Kjeldahl apparatus (Model UDK 126 D, VELP Scientifica Srl, Usmate Velate, Italy) with adjustable heaters for individual flasks. For this assay, 1 g of grounded dry sample (ca 0.7–1 mm) mixed with 20 mL of deionized water in a Kjeldahl tube was digested by adding H_2_SO_4_ (12 mL), K_2_SO_4_ (7.0 g), and CuSO_4_ (0.8 g). If the mixture foamed, 30–35% H_2_O_2_ (3 mL) was slowly added. The nitrogen in the sample was converted to nonvolatile ammonium sulphate. The digest was cooled down by adding distilled water (80 mL); then, the ammonium sulphate was converted to volatile ammonia gas by heating with 40% NaOH. The ammonia was steam-distilled into an excess of a boric acid solution, where it was trapped by forming ammonium borate. The amount of H_3_BO_3_-receiving solution formed was determined by titration using standard 0.10 M HCl to a violet endpoint. To obtain the crude protein content of raw materials, the N content was multiplied by a conversion factor that reflects the percentage of N in the sample protein. The crude protein content was expressed as g of Kjeldahl nitrogen per g of dry sample (g/g_DM_).

Total fat was quantified by gravimetry after extraction (AOAC Method 920.39) [51] using the Randall modification of the standard Soxhlet method to reduce the time needed for the extraction process. Briefly, 2 g of grounded dry sample (ca 0.7–1 mm) was mixed with C_2_H_6_O (2 mL) and HCl (100 mL). The extraction system was then kept under gentle agitation at 80 °C for 30 min. After this stage, a miscible solution of petroleum ether (25 mL) and diethyl ether (25 mL), both of which are solvents, was added. After extraction, the solvent was released with an R-200/205 Rotavapor (BÜCHI Labortechnik AG, Flawil, Switzerland), and the last traces were removed by placing the flask with the extract in a heater at 80 °C overnight. Lipid contents were gravimetrically determined from the difference in the weight of samples before and after drying, and it was expressed as g of lipids per g of dry sample (g/g_DM_).

### 3.4. Conventional Solid/Liquid Extraction (SLE)

The process of the conventional SLE of agri-food residues was carried out using an orbital incubator (Stuart SI50, Salford Scientific Supplies Ltd., Salford, Manchester, England) at 25 °C and 160 rpm. Raw material (20 g) was mixed with 200 mL of the solvent (methanol, ethanol, and acetone) at different dilutions with distilled water (ranging from 20 to 100% *v*/*v*) and kept at room temperature for up to 24 h. Then, the extract was centrifuged (PK121R model, ALC International, Cologno Monzese, Milan, Italy) at 10,000× *g* at 4 °C for 10 min to obtain clear supernatants, which were used for further qualitative and quantitative analyses. Extractions were performed in duplicate.

The response surface methodology (RSM) model for the design of experiments (DOE) was used to model and optimize the conventional solid/liquid extraction process. A factorial experiment involving two factors (the type of solvent, discrete numeric factor, and solvent concentration) was used to optimize the polyphenol extraction yield. The I-optimal design was applied in this study using the Design-Expert 11.1.2.0 software (Statease Inc., Minneapolis, MN, USA) to determine the number of experiments to be evaluated to optimize the two independent variables and the response variable. The independent variables used in the RSM design are listed in Table 3.

The experiments with two replications were fitted with a second-order polynomial, as seen in Equation (1).
(1)$$Y=\beta_{0}+\beta_{1}X+\beta_{2}X^{2}$$

In this equation, the value of the total phenols, which is used as the response variable (Y), is related to the experimental factor (X). The coefficients of the polynomial model correspond to the constant term (β_0_), linear effects (β_1_), and pure second-order interaction effects (β_2_) obtained by the multiple nonlinear regression method. The performance and adequacy of the developed RSM model were assessed following the statistical parameters estimated using the analysis of variance (ANOVA) technique and included the determination coefficients (R^2^, Adj R^2^), the coefficient of variation (C.V.), and probability values (*p*-value).

### 3.5. HPH-Assisted Extraction

The HPH-assisted extraction experiments were performed by first mixing the raw materials suspended in distilled water at 5 wt% using a high-shear mixer (HSM) (Ultra Turrax T-25, IKA Labortechnik, Staufen, Germany) equipped with an S25-N18 G rotor and operated at 20,000 rpm in an ice bath to avoid any increase in temperature. After 5 min of processing, the HSM suspensions were submitted to HPH processing using a unit developed in-house equipped with an orifice valve (model WS1973, Maximator JET GmbH, Schweinfurt, Germany) of 200 μm at 80 MPa for up to 20 equivalent passes. After each pass, the suspensions were cooled down in a tube-in-tube heat exchanger set at different temperatures to ensure that the product temperature ranged between 5 and 50 °C. Then, the extract was centrifuged (PK121R model, ALC International, Cologno Monzese, Milan, Italy) at 10,000× *g* at 4 °C for 10 min to obtain clear supernatants, which were used for further qualitative and quantitative analyses. Extractions were performed in duplicate.

### 3.6. Qualitative and Quantitative Analyses of Extracts

#### 3.6.1. Bioactivity Determination

Total phenols were quantified through the Folin–Ciocalteau assay [52,53], which was adopted to study the natural antioxidant compounds; it measures the capacity of a compound to reduce Folin’s reagent, thus giving an estimation of its antioxidant capacity. Briefly, a sample of extract or liquor (1 mL opportunely diluted with distilled water) was added to 10% *v*/*v* Folin–Ciocalteau’s reagent (5 mL) and allowed to stand for 5 min at room temperature. Then, 7.5 wt% sodium carbonate solution (4 mL) was added to the mixture. After vortexing for 1 min, the mixture was incubated at room temperature for 60 min in darkness. The absorbance of the reacting mixture was then read at 765 nm using a V-650 ultraviolet–visible (UV-Vis) spectrophotometer (Jasco Inc., Easton, MD, USA). Total phenols were expressed as milligrams of gallic acid equivalents per gram of dry samples (mg_GAE_/g_DM_) by means of a calibration curve obtained with a gallic acid standard at different concentrations (10–100 mg/L).

The reducing capacity of extracts was evaluated with a FRAP (ferric-reducing antioxidant power) assay carried out according to the method described by Benzie and Strain (1996) [54] with some modifications. The FRAP working solution was prepared by mixing 0.3 M sodium acetate buffer, 10 mM TPTZ (2,4,6-Tris (2-pyridyl)-s-triazine) solution in 40 mM HCl, and 20 mM ferric solution at a ratio of 10:1:1 (*v*/*v*/*v*). The freshly prepared FRAP reagent solution (2.5 mL) was mixed with an opportunely diluted sample (0.5 mL) and incubated for 10 min at ambient temperature. The change in absorbance due to the reduction of the ferric-tripyridyltriazine (Fe III-TPTZ) complex by the antioxidants present in the samples was measured at 593 nm using a V-650 UV-Vis spectrophotometer against a blank (applying the same analysis conditions). Ascorbic acid was used as the standard for the calibration curve, and the FRAP values were expressed as ascorbic acid equivalents present in the dried sample (mg_AA_/g_DM_).

Furthermore, the antioxidant activity was measured by a 1,1-diphenyl-2-picryl hydrazyl (DPPH) spectrophotometric assay using the method reported by Rapisarda et al. (1999) [55]. A 25 ppm DPPH solution in methanol was used as the stock solution. The DPPH solution (3.9 mL) was mixed with extracts (0.1 mL) and kept in complete darkness for 5 min. The absorbance was therefore determined at 517 nm. The following formula (Equation (2)) was used to compute the percentage of inhibition:(2)$$\%\;\mathrm{inhibition}=\frac{{\mathrm{Abs}}_{\mathrm{stock}}-{\mathrm{Abs}}_{\mathrm{sample}}}{{\mathrm{Abs}}_{\mathrm{stock}}}\cdot 100$$

The identification of chlorogenic acid, caffeic acid, gallic acid, quercetin 3-glucoside, epicatechin, and rutin contained in the extracts was carried out by high-performance liquid chromatographic (HPLC-DAD) analysis. The HPLC analysis to determine the relative concentration of single phenolic compounds was performed in a Waters 1525 Separation Module coupled to a Waters 2996 photodiode array detector (Waters Corporation, Milford, MA, USA). Before the injection, all the collected extracts were filtered with 0.45 μm filters. The mobile phase for the determination of the relative concentration of phenols consisted of phosphoric acid (0.1%, eluent A) and methanol (100%, eluent B). The gradient elution program was as follows: 0–30 min from 5% B to 80% B; 30–33 min 80% B; and 33–35 min from 80% B to 5% B. The flow rate of the mobile phase and the injection volume was, respectively, 0.8 mL/min and 5 µL. Chromatograms were acquired at the fixed wavelength by comparing the HPLC retention time and visible absorption spectra with those of the commercial standard.

#### 3.6.2. Morphological Properties

A qualitative determination of the structural properties of micronized agri-food residues through different treatments was performed using an optical microscope (Nikon Eclipse TE 2000S, Nikon Instruments, Europe B.V., Amsterdam, The Netherlands) with a 10× objective coupled to a DS Camera Control Unit (DS-5M-L1, Nikon Instruments Europe B.V, Amsterdam, The Netherlands) for image acquisition and analysis.

#### 3.6.3. Particle Size Distribution

The size distributions (PSD) of micronized agri-food residue particles in aqueous suspensions were measured by laser diffraction at 25 °C with a Mastersizer 2000 instrument (Malvern instrument Ltd., Malvern, UK) using the Fraunhofer approximation, which does not require knowledge of the optical properties of the sample, as reported by Pirozzi et al. [44]. The temperature of the cell was maintained at 25 ± 0.5 °C, and an average of triplicates was determined. The characteristic diameters d(0.1), d(0.5), and d(0.9) were evaluated, corresponding to the 10th, 50th (median value), and 90th percentile of the cumulative size distribution of the suspensions.

### 3.7. Statistical Analysis

All experiments and analyses were performed in triplicate on independently prepared samples unless otherwise specified, and the mean and standard deviation (SD) of the experimental values were calculated. Statistically significant differences (*p* ≤ 0.05) among the mean values were assessed by one-way ANOVA and Tukey’s test (*p* < 0.05) using the SPSS statistical software (version 20, SPSS Inc., Chicago, IL, USA).

## 4. Conclusions

This study explores innovative and sustainable extraction techniques to recover antioxidant compounds from agri-food residues. The process of the conventional solid–liquid extraction (SLE) of phenolic compounds was optimized through response surface methodology for various agri-food residues, considering the type and concentration of the organic solvent in water. The yields of total phenols and the antioxidant activity obtained from the optimized SLE conditions served as a benchmark to evaluate the potential of high-pressure homogenization (HPH) treatment (80 MPa for 20 equivalent passes at 25 °C) to enhance the extraction yields using pure water as the solvent. The results demonstrated a notable increase (~20%) in the yield of phenolic compounds and the associated antioxidant capacity compared to SLE for all the residues that were suitable to be treated by HPH (roasted coffee beans, wheat middlings, wheat bran, and tomato pomace). The particle size reduction achieved through HPH treatment (a reduction in d(0.5) higher than 90% for roasted coffee beans, wheat middlings, wheat bran, and tomato pomace) significantly increased the specific surface area, thereby improving the mass transfer rate and extractability of polyphenols.

The comparison between the SLE and HPH treatments highlights the advantages of the latter in terms of its efficiency, sustainability, and product quality, making it a promising green, eco-friendly and cost-effective technique for recovering bioactive compounds from agri-food residues derived from industrial transformation. HPH treatment reduces extraction time, eliminates the need for solvents with negative economic and environmental impacts, and operates at lower processing temperatures, preventing the degradation of temperature-sensitive compounds. These practical aspects of agri-food residue valorization through HPH-assisted extraction are valuable for industrial applications, including as food supplements or cosmetics formulations. Furthermore, the reduction in the economic and environmental footprints associated with HPH-assisted extraction aligns with the broader goal of promoting circular economy principles and reducing waste. Further research is necessary to fully understand the relationship between the composition of the extracts and their functional properties. Additionally, assessing the feasibility of scaling up the HPH-assisted extraction process and conducting economic evaluations will be crucial for its successful implementation in large-scale industrial applications.

## Figures and Tables

**Figure 1 molecules-28-05657-f001:**
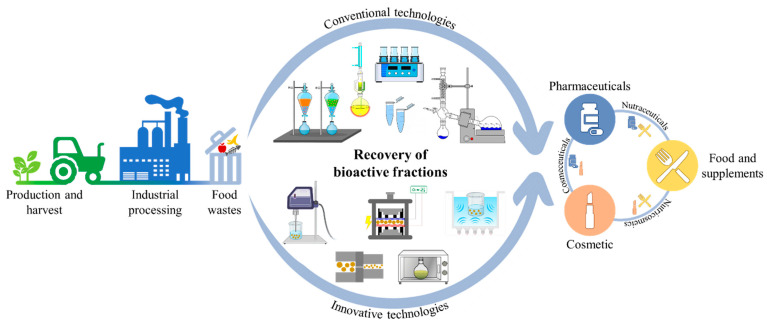
Recovery of bioactive fractions from agri-food residues by conventional and innovative technologies for different applications.

**Figure 2 molecules-28-05657-f002:**
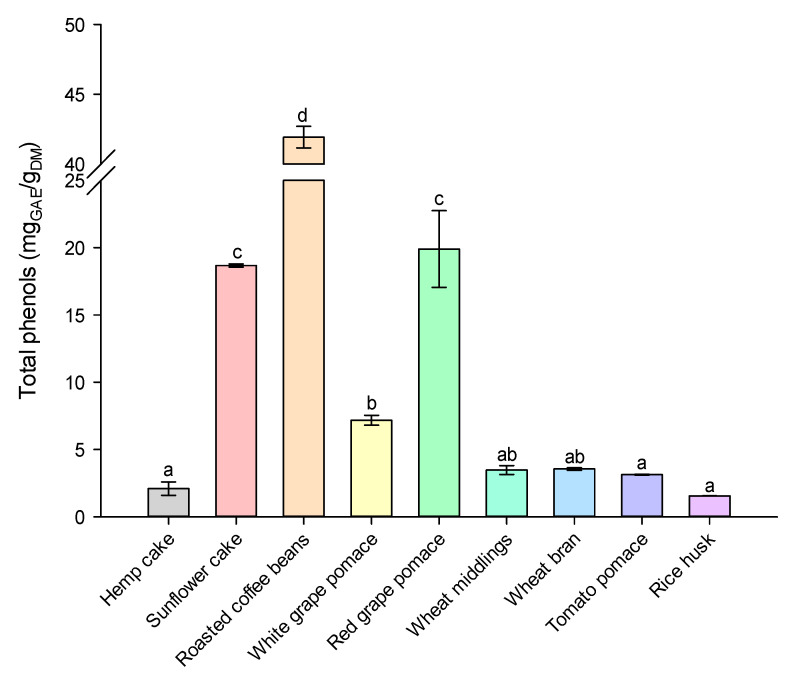
Total phenol content of extracts obtained after SLE extraction at 25 °C from agri-food residues after 24 h of diffusion. Different letters above the bars indicate significant differences among the mean values (*p* < 0.05).

**Figure 3 molecules-28-05657-f003:**
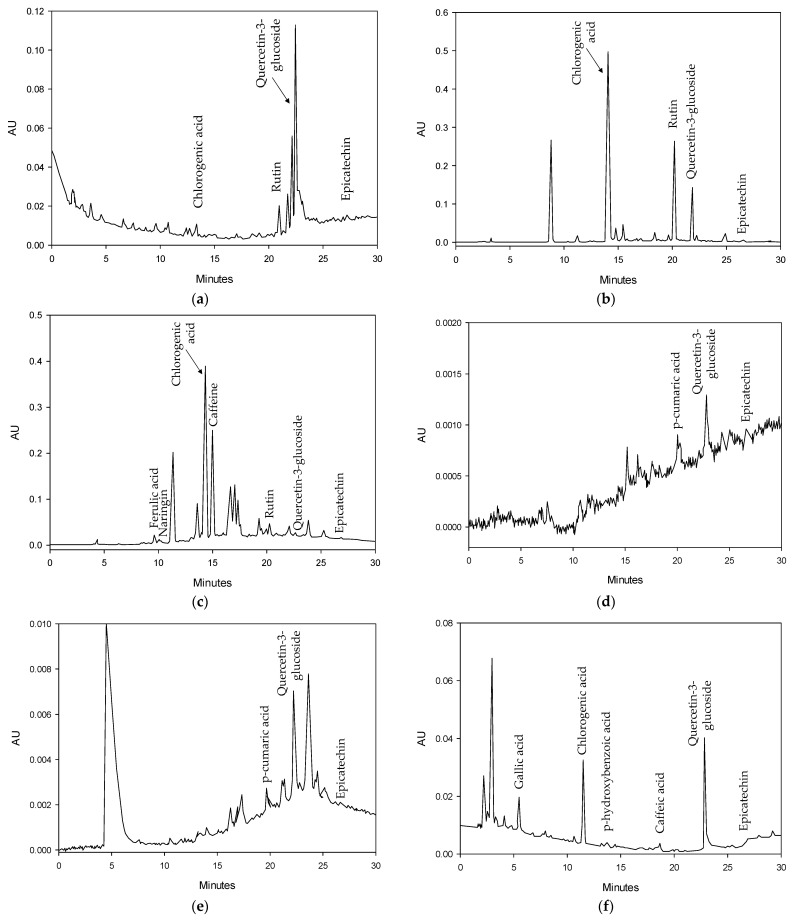
HPLC-PDA chromatograms of extracts obtained after 24 h of SLE at 25 °C from (**a**) hemp cake with 60 v% methanol; (**b**) sunflower cake with 60 v% ethanol; (**c**) roasted coffee beans with 40 v% acetone; (**d**) white grape pomace with 20 v% acetone; (**e**) red grape pomace with 60 v% acetone; (**f**) wheat middlings with water; (**g**) wheat bran with water; (**h**) tomato pomace with 80 v% acetone; and (**i**) rice husk with 50 v% ethanol.

**Figure 4 molecules-28-05657-f004:**
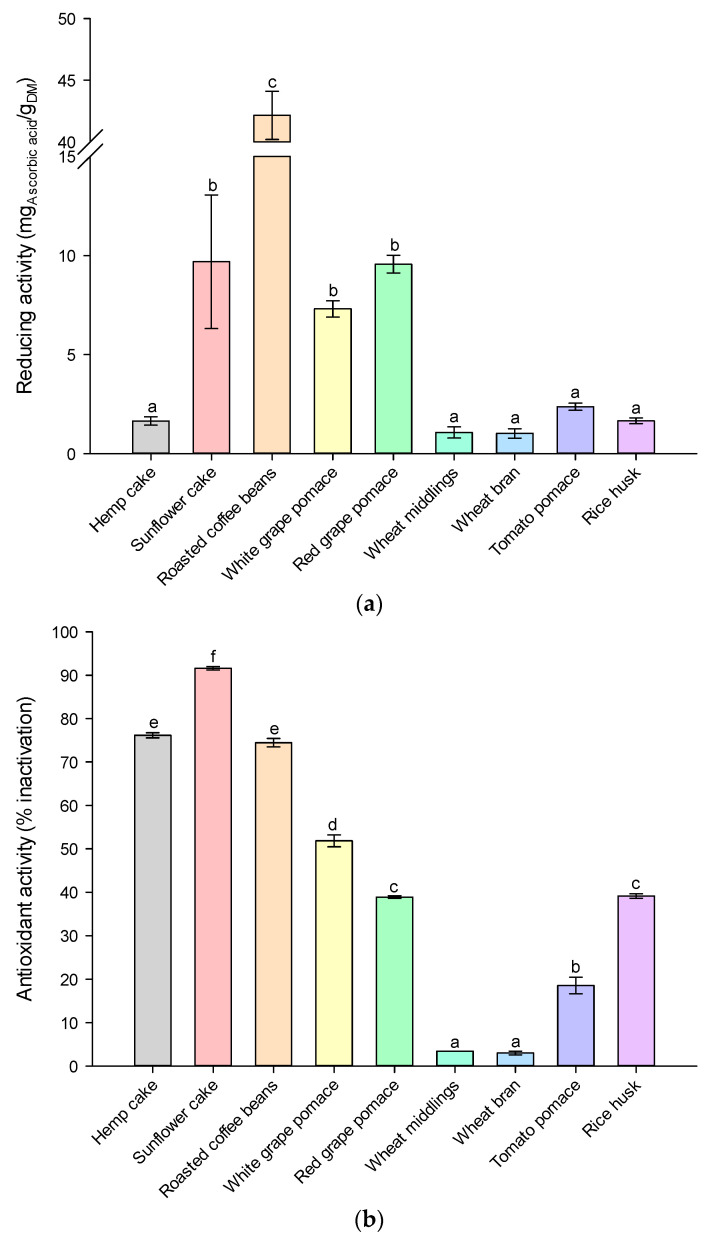
Antioxidant activity of extracts obtained after SLE extraction at 25 °C from agri-food residues after 24 h of diffusion: (**a**) FRAP and (**b**) DPPH assays. Different letters above the bars indicate significant differences among the mean values (*p* < 0.05).

**Figure 5 molecules-28-05657-f005:**
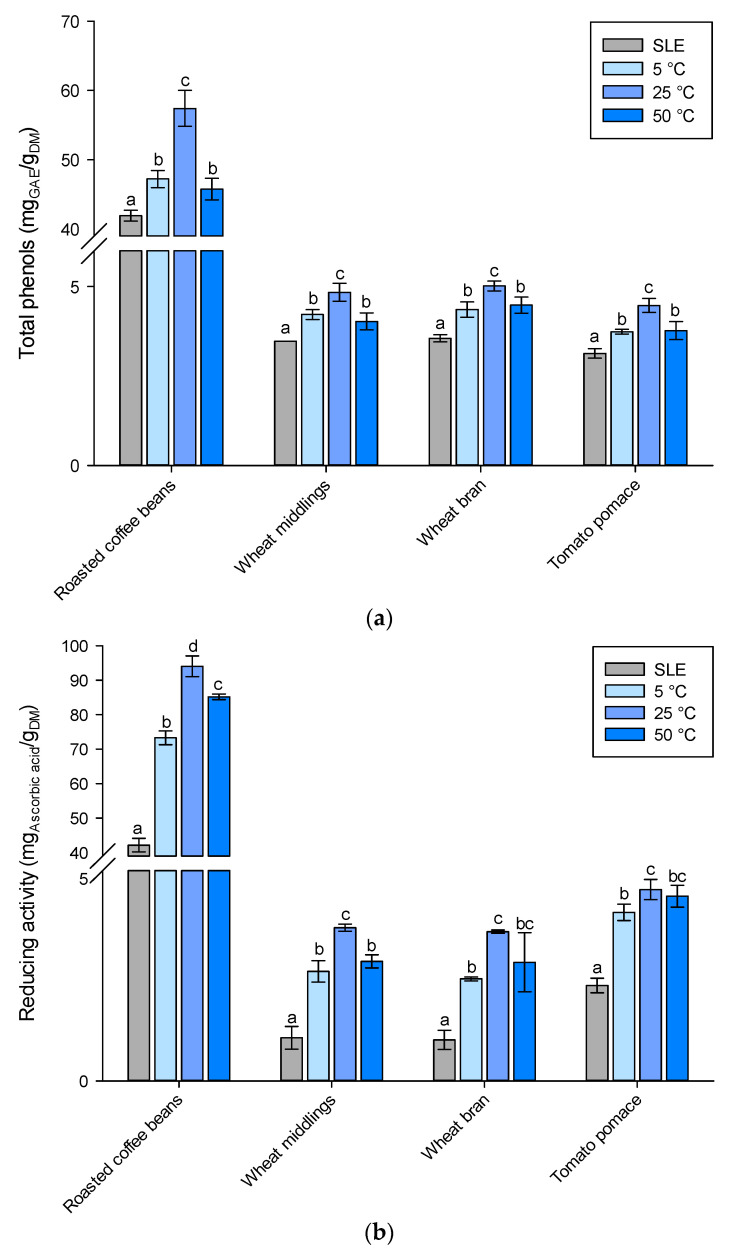
(**a**) Total phenols and (**b**) reducing activity of agri-food residue extracts obtained through (
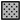
) SLE extraction with organic solvent at optimum operating conditions (reported in Appendix A) and HPH-assisted extraction in water at (
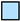
) 5 °C, (
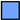
) 25 °C, and (
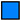
) 50 °C. Different letters above the bars indicate significant differences among the mean values (*p* < 0.05).

**Figure 6 molecules-28-05657-f006:**
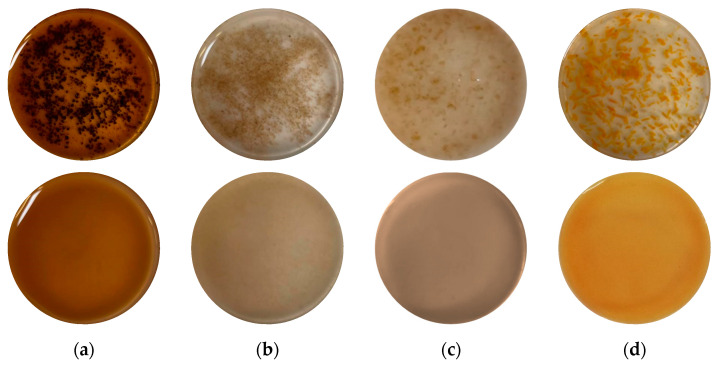
Visual observations of (**a**) roasted coffee beans; (**b**) wheat middlings; (**c**) wheat bran; and (**d**) tomato pomace in water suspension. First row indicates agri-food residue suspensions after HSM treatment; second row indicates agri-food residue suspensions after HPH treatment.

**Figure 7 molecules-28-05657-f007:**
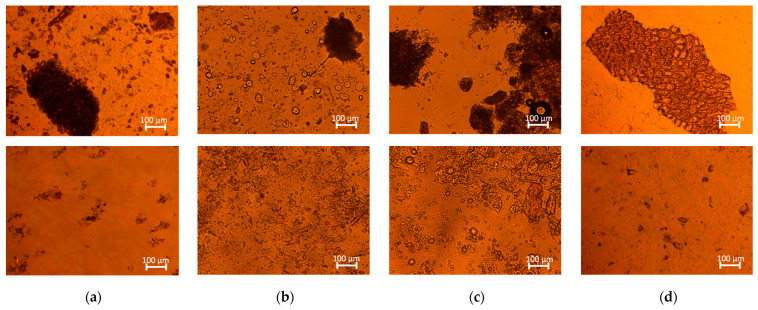
Brightfield micrographs of (**a**) roasted coffee beans; (**b**) wheat middlings; (**c**) wheat bran; and (**d**) tomato pomace in water suspension. First row indicates agri-food residue suspensions after HSM treatment; second row indicates agri-food residue suspensions after HPH treatment.

**Figure 8 molecules-28-05657-f008:**
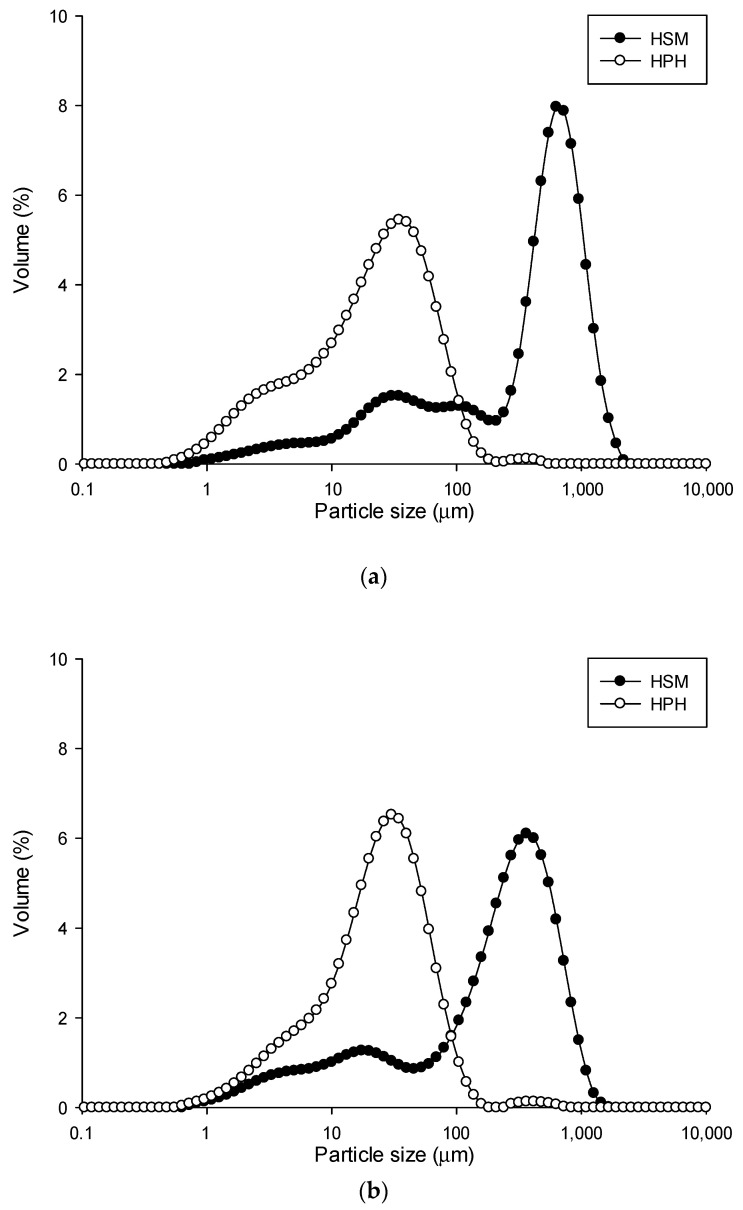
Particle size distribution of (**a**) roasted coffee beans; (**b**) wheat middlings; (**c**) wheat bran; and (**d**) tomato pomace in water suspension treated by (
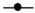
) HSM and (
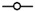
) HPH.

**Table 1 molecules-28-05657-t001:** Chemical composition of agri-food residues reported as relative percentage on dry basis (except for moisture).

	Moisture (%)	Ash (%)	Protein (%)	Fat (%)	Carbohydrates (%)
Hemp cake	7.99 ± 0.06	6.40 ± 0.04	24.78 ± 0.49	5.30 ± 0.09	63.52 ± 0.49
Sunflower cake	10.75 ± 0.06	5.93 ± 0.03	24.30 ± 1.24	2.30 ± 0.21	68.47 ± 1.24
Roasted coffee beans	5.97 ± 0.04	5.10 ± 0.81	16.96 ± 0.41	1.00 ± 0.02	76.94 ± 0.91
White grape pomace	80.41 ± 1.99	10.52 ± 3.08	58.13 ± 7.95	1.55 ± 0.14	29.80 ± 9.84
Red grape pomace	62.58 ± 2.21	35.88 ± 5.80	51.88 ± 1.77	2.70 ± 0.33	9.54 ± 3.55
Wheat middlings	10.59 ± 0.01	3.79 ± 0.25	18.53 ± 1.21	0.79 ± 0.06	76.90 ± 1.23
Wheat bran	11.54 ± 0.10	5.83 ± 0.60	19.10 ± 0.40	0.85 ± 0.10	74.22 ± 0.72
Tomato pomace	80.70 ± 0.83	4.90 ± 0.27	14.65 ± 0.21	1.20 ± 0.14	79.25 ± 0.34
Rice husk	6.72 ± 0.13	18.71 ± 0.23	2.56 ± 0.25	0.82 ± 0.15	76.44 ± 0.34

**Table 2 molecules-28-05657-t002:** Characteristic diameters (μm) of the particle size distribution of the aqueous suspensions of agri-food residue treated by HSM and HPH.

	Treatment	d(0.1)	d(0.5)	d(0.9)	D[4,3]	D[3,2]
Roasted coffee beans	HSM	1.83 ± 0.14	451.45 ± 16.90	978.34 ± 18.41	468.85 ± 16.74	34.51 ± 2.13
	HPH	2.75 ± 0.01	21.31 ± 0.09	65.03 ± 0.30	29.90 ± 0.42	7.53 ± 0.05
Wheat middlings	HSM	8.55 ± 0.13	220.64 ± 3.02	595.37 ± 8.92	266.08 ± 5.17	21.80 ± 0.31
	HPH	4.40 ± 0.05	22.62 ± 0.25	59.55 ± 0.48	30.68 ± 0.36	10.43 ± 0.14
Wheat bran	HSM	36.28 ± 2.08	374.70 ± 4.63	898.21 ± 6.77	438.26 ± 3.54	52.85 ± 1.17
	HPH	5.04 ± 0.01	34.41 ± 0.11	97.22 ± 1.30	47.30 ± 1.20	13.23 ± 0.03
Tomato pomace	HSM	67.20 ± 0.89	349.43 ± 6.29	1089.82 ± 51.78	473.18 ± 16.09	113.74 ± 1.97
	HPH	7.78 ± 0.14	29.97 ± 0.25	78.22 ± 0.50	38.47 ± 1.06	14.47 ± 0.16

**Table 3 molecules-28-05657-t003:** The I-optimal design used with the indication of the actual value of the two variables.

Run Number	Type of Solvent	Solvent–Water Mixture Ratios (% *v*/*v*)
1	Acetone (1)	31
2	Acetone (1)	100
3	Ethanol (2)	20
4	Ethanol (2)	65
5	Ethanol (2)	59
6	Ethanol (2)	100
7	Ethanol (2)	20
8	Ethanol (2)	66
9	Methanol (3)	20
10	Methanol (3)	100
11	Methanol (3)	57

## Data Availability

The data are contained within the article.

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
