# Peer review of "Impact of High-Pressure Homogenization on Enhancing the Extractability of Phytochemicals from Agri-Food Residues"

_molecules, 2023, doi:10.3390/molecules28155657_

Round 1
Reviewer 1 Report
The paper is interesting to read, contains reports about “Enhancing the recovery of phytochemicals from agri-food residues via high-pressure homogenization”. However, the structure of the manuscript is not well laid out, some major points should be taken into account and corrected:
Title: Well-deserved But the following title is suggested
Impact of High-Pressure Homogenization on Enhancing the Extractability of phytochemicals from agri-food residues
Abstract:
the word "novel" such claims needs proof. Consider explaining novelty in some other way. Some helpful words are "innovative”, "cutting-edge", "alternative", "previously unknown".
The word "highly" rarely highly contributes to better understanding. Consider removing it or, if important quantifying it.
Replace "utilization" with simple "use".
The word "significantly" is often significantly misused and vague. It might mean statistically significant or significant to the author. State significance quantitatively, e.g. "increased by 42%". Other alternatives: "substantially, notably"
The sentences seem to be too long. Consider shortening or splitting it.
Keywords: It should be written like this
high-pressure homogenization; Extraction, Bioactive compounds Agri-food waste
Introduction:
high-pressure homogenization is TREATMENT or pre- treatment???
The influencing factors on high-pressure homogenization have not been investigated
You can use these articles
Effects of high-pressure homogenization extraction on the physicochemical properties and antioxidant activity of large-leaf yellow tea polysaccharide conjugates https://doi.org/10.1016/j.procbio.2022.09.020
Effects of high-pressure homogenization treatment on physiochemical properties of novel plant proteins https://doi.org/10.1016/j.afres.2023.100285
Sucrose ester alleviates the agglomeration behavior of bamboo shoot dietary fiber treated via high pressure homogenization: Influence on physicochemical, rheological, and structural properties https://doi.org/10.1016/j.foodchem.2023.135609
Lines 25 to 39 are a bit long about waste, please be more concise
It is better to be mentioned to the other methods for the recovery of phytochemicals from agri-food residues like:
Optimizing extraction of berberine and antioxidant compounds from barberry by maceration and pulsed electric field-assisted method
10.3233/JBR-200603
The Introduction section of the paper should give greater context to the project. You could write more expressively and clearly
The study aims have been answered.
The abbreviation AFR occurs only a few times. Since abbreviations are hard to decrypt, just spell it out each time. It is easier to read a few words than to search for meanings of abbreviations.
Line 25. Verify that the verb "generate" really describes a generation process. Otherwise, consider replacing it with "cause".
Line 34. Replace "utilize" with simple "use".
Line 38. Consider replacing "subsequent" with consequent
English should be thoroughly revised. Grammatical and vocabulary errors appear with a specific frequency. Some sentences need to be understood.
Consider if your readers know the Latin expressions "e.g.,". It might be better to write "for example".
Line 53. Try to say it without "and/or" monstrosity. Often, just "and" or "or" is enough.
Line 71-74 There is no need to explain the response surface methodology (RSM) because it is a well-known method.
Materials and Methods:
Relevant references are not well cited throughout the methods
In lines 479-489 Determine the concentrations.
The protocol of Total polyphenol content (TPC) was incomplete You can follow this protocol:
“Total polyphenol content (TPC) in extracts was determined by Folin–Ciocalteu colorimetric method according to Azghandi Fardaghiet al. (2021).
Results and Discussion
It would not be better to extract fiber, extract caffeine, extract alkaloids, and extract polyphenols separately from Hemp cake, Sunflower cake, Roasted coffee beans, White grape pomace, red grape pomace, Wheat middlings, Wheat bran, Tomato pomace, Rice husk???
The resolution & quality of Fig. Should improve
You did not use meaningful alphabets).Significant letters that indicate statistically significant differences have not been used.
Predicted and experimental values of the response variables at optimum extraction not done
Model optimization and verification was not done
The efficiency of the model was not checked) F-value and p-value (
Conclusions:
Conclusions are far too general and vague. These need to be precise and explain how the aims of the paper have been achieved while emphasizing the originality of the work.
References:
They were completely relevant and new but lines 644,650,647,639,619,614,590, If possible, update them.
-
Reviewer 2 Report
The authors investigated the effect of high-pressure homogenization on the recovery of phytochemicals from 9 agri-food residues including hemp cake, sunflower cake, roasted coffee beans, white grape pomace, red grape pomace, wheat middlings, wheat bran, tomato pomace, and rice husk. I have thoroughly read the paper and I don’t think the authors tell a good scientific story to the readers of Molecules. I recommend major revision.
Some issues are listed below:
1. Title, please directly tell the readers the research objectives (nine or four agri-food residues).
2. Abstract, the background is too long, one or two sentences for the background of the study are enough.
3. Section 2.2, the authors spent too much time to optimize the conventional extraction process of phytochemicals from 9 agri-food residues based on RSM. According to the paper, the main aim of the study is the use of high-pressure homogenization on the phytochemicals extraction from 9 agri-food residues; so, I think RSM analysis is pre-experiment and should be move to the supplementary files.
4. Section 2.3.1, the authors only investigated 4 agri-food residues including roasted coffee beans, wheat middlings, wheat bran, and tomato pomace, why?
5. Section 2.3.2, the authors investigated the effect of high-speed mixing (HSM) on the on physical characteristics of AFRs suspensions. I am confused about this. What do the authors want to tell the readers? Is this the topic of the study? If the authors do want to show this result, please add some related knowledge in the Introduction section and reconsider the title of the study.
6. As a total, I think the paper is a patchwork without any design.
Editing errors:
Line 29, delete “including”.
Fig. 1, improve the figure quality.
Line 54, “and t heat treatments”, please check.
Line 204, DAD or PDA, please check; according to 3.6.1, I think PDA is used in the study.
Line 240, sole solvent, period (.) is missing.
Reviewer 3 Report
The manuscript entitled: "Enhancing the recovery of phytochemicals from agri-food residues via high-pressure homogenization" is about the application of a novel method to improve phytochemical extraction from agri-foo residue. In general, the research is interesting, well designed and the objectives aligned with the journal's aims and scopes. There are some comments below to improve the quality of the manuscript before final decision by the editor:
1- Title: the title is so general and better to make it more specific
2- Abstract: Improve the abstract by incorporation some quantitative data in the results. Also, recommend using short sentences to make it easier to read for all readers.
3- Introduction: Almost good structure, please check recent references and try to update the references.
4- Results and discussion: The presented figures and tables are good and the results are justified clearly.
5- Materials and Methods: Recommend making them brief and informative. All methods should have a proper reference(s) from the standard method or a published article.
6- References: Although all references are related to the research but better to replace some with recent years, especially after 2020.
Good Luck
Reviewer 4 Report
The presentation of this manuscript was good enough in terms of presentation and methodology. Moreover, the paper is subjected to major improvement.
1. Specify the lipid content in table 1.
2. Define the difference between conventional extraction and SLE.
3. What are the processes variables considered in RSM?
4. How did the solvent types were considered in the regression equation? In figure 2, there are very smooth response with changing the types of solvent – I am doubt on similar response in reality.
5. How did the predominate phenolic acids (chlorogenic acid and quercetin-3) effect the extraction efficiency?
6. Evaluate the extraction efficiency for different source based on total phenolic acid.
7. Make some comparison of FRAP and antioxidant activities with literature values.
8. Figure 7 is not clear. Define the difference clearly between SLE (which organic solvent/temperature) and other temperature effect.
Moderate editing of English language required
Reviewer 5 Report
Comment 1: The abstract should state briefly the purpose of the research, the principal results and major conclusions. An abstract is often presented separately from the article, so it must be able to stand alone.
Comment 2: Especially, the introduction section needs to re-organize. The major debate or Argument is not clear stated in the introduction session. Hence, the contribution debates are weak in this manuscript. I would suggest the author to enhance your literature discussion and arrives your debate or argument.
Comment 3: The presented paragraph in page 2 “This study explores an alternative approach involving solvent-free extraction methods. In this context, high-pressure homogenization (HPH) technology has gained increasing interest as a nonthermal and easily scalable cell disruption technique for plant tissues. HPH induces the rupture of cell envelops, enhancing the extractability of target intracellular compounds from food by-products matrices. Importantly, HPH achieve this without the need for organic solvents, ensuring the safety of the final products intended for use in food supplements, pharmaceutical and cosmeceutical applications. However, limited studies have demonstrated the feasibility of the HPH in enhancing the extractability of phenolic compounds from AFRs.” needs to be re-written. It is too general.
Comment 4: The quality of the figure 5 “HPLC-PDA chromatograms of extracts obtained after 24 h of SLE at 25 °C from: (a) hemp cake with 60 v% methanol; (b) sunflower cake with 60 v% ethanol…” is not acceptable.
Comment 5: Please explain your results into steps and links to your proposed method.
Comment 6: It is suggested to add articles entitled “Ratnawati et al. Response Surface Methodology for Formulating PVA/Starch/Lignin Biodegradable Plastic”, “Narala et al. Acid Whey Valorization for Biotechnological Lactobionic Acid Bio-production” and “Grinberga-Zalite & Zvirbule. Analysis of Waste Minimization Challenges to European Food Production Enterprises” to the literature review.
Comment 7: I would like to request the author to emphasis on the contributions on practically and academically in implication session.
Comment 8: It is mentioned in the results section that: This confirmed that a strong positive correlation was found between total phenols and antioxidant activity, with a Pearson correlation coefficient of about 0.954, indicating that the overall antioxidant activity of the extracts is mostly associated with the phenolic compounds. The interpretation of this section seems to be weak, the authors should develop this section.
Comment 9: Please make sure your conclusions' section underscore the scientific value added of your paper, and/or the applicability of your findings/results, as indicated previously. Please revise your conclusion part into more details. Basically, you should enhance your contributions, limitations, underscore the scientific value added of your paper, and/or the applicability of your findings/results and future study in this session.
Round 2
Reviewer 1 Report
The manuscript can be accepted in its present form.
Reviewer 2 Report
Accept as it is
Reviewer 4 Report
Accept in present form
Minor editing of English language required
Reviewer 5 Report
The authors have successfully addressed the reviewer comments. Well done. Several points to consider to improve quality of this paper.